# Determinants of the Economic Vulnerability of Businesses to Pandemics and Similar Events

Clement A. Tisdell

School of Economics, The University of Queensland, Brisbane 4072, Australia; c.tisdell@uq.edu.au;
Tel.: +61-7-3365-6306

**Abstract:** After providing a general overview of factors that make businesses economically vulnerable to pandemics (such as COVID-19), this article identifies specific elements that increase the vulnerability of businesses to pandemics. These specifics include the extent to which the demand for their production declines, how easy it is for them to reduce the costs of their production (cost escapability), the importance of disruptions or breaks in the supply chains of inputs utilized by businesses, and their ability to sustain their liquidity. Businesses that rely on personal contacts for sales are especially threatened, for example, those in the hospitality and tourism sector. However, others are also vulnerable for the reasons given. Nevertheless, some businesses do gain as a result of pandemics and similar events. Their economic gain adds to GDP. However, it could be more appropriate to regard their gains as a part of the cost of a pandemic rather than a benefit of it. The effect on the vulnerability of businesses if government policies designed to control pandemics is also considered. The main original contribution of this article is to show how the microeconomic theory of the firm can be adapted to conceptualize the vulnerability of individual businesses to pandemics, particularly COVID-19, while also noting the limitations of this approach.

**Keywords:** COVID-19; demand volatility; epidemics; inescapable costs; insolvency; labour problems; product chains; public policy and pandemics

## 1. Introduction

Bloom et al. (2018) state that new and resurgent infectious diseases often have far-reaching adverse economic effects. They point out that:

'Beyond shocks to the health sector, epidemics force both the ill and their caretakers to miss work or to be less effective at their jobs, driving down and disrupting productivity. Fear of infection can result in social distancing or closed schools, enterprises, commercial establishments, transportation and public services—all of which disrupt economic and other socially valuable services'.

The novel coronavirus, or COVID-19, has had all of these effects and more. However, the nature and consequences of different types of pandemics and infectious diseases varies considerably (Tisdell 2020). This influences their financial risks to businesses (Hassan et al. 2020). Furthermore, the attributes of individual businesses (for example, their cost and demand characteristics) influence their economic vulnerability to these events.

The main purpose of this article is to systematically identify those factors that render individual businesses financially vulnerable to pandemics, especially COVID-19, in an integrated and general manner. Its conceptual/theoretical nature is mostly derived from the principles of microeconomics. While some recent articles assess the ability of individual firms to remain financially viable given the occurrence of epidemics (particularly COVID-19), they do not do so in the same original manner as in this article. For example, Obrenovic et al. (2020) consider an entirely different set of factors in assessing the ability of an enterprise to sustain its operations and productivity during the COVID-19 pandemic. Bartik et al. (2020) conducted a survey of the impact of COVID-19 on small businesses in the USA

but do not provide a conceptual analysis of their financial vulnerability to the presence of COVID-19. Hassan et al. (2020) provide a useful economic overview of factors influencing the firm-level financial exposure to epidemic diseases (see especially their Appendix Table A8). However, they do not provide the same type of theoretical exposition as that set out in this article.

Note that the theoretical content of this article is backed up by empirical observations drawn from the relevant literature. It should also be noted in advance that it is not intended to deal with the macroeconomic consequences of pandemics, especially COVID-19. These are considered, for example, in Eichenbaum et al. (2020); Saif et al. (2021); Vidya and Prabheesh (2020); Padhan and Prabheesh (2021).

This article identifies differences in the economic vulnerability of businesses to pandemics and infectious diseases, paying particular attention to COVID-19. This subject is discussed in the following order. (1) The consequences of differences in the nature of pandemics and infectious diseases for vulnerability of businesses and their survival. (2) The role of governments in moderating or adding to this vulnerability is then considered. The economic characteristics of firms that influence their risks are subsequently identified. These include (3) demand side risks; (4) risks due to lack of the availability of factors of production, e.g., supply chain disruptions and labor shortages; (5) variations in the nature of cost structures, e.g., the extent to which costs are escapable; as well as (6) liquidity considerations. This exposition is followed by findings and a discussion which, among other things, identifies possible business losers and winners from the occurrence of infectious diseases, and a conclusion.

## 2. Methodology

This article draws primarily on microeconomic analysis and the relevant literature to provide a theory or conceptualization of the factors that increase the financial vulnerability of businesses to pandemics (particularly COVID-19) and similar events. This conceptualization and theoretical analysis is strengthened by the provision of examples drawn from the scholarly literature and from elsewhere. This appears to be the first time that this type of analysis and conceptualization has been completed. Sections 3–8 contain the analytical content of the article and Section 9 summarizes and discusses the main results.

## 3. Differences in the Nature of Pandemics and Infectious Diseases—Their Consequences for the Economic Vulnerability of Business

The occurrence of new infectious diseases and their attributes are very difficult to predict. This has been illustrated by the emergence of COVID-19 (Ma et al. 2020). It is, therefore, difficult for businesses to prepare in advance for evolution of these diseases. Furthermore, the economic consequences of new infectious diseases take time to determine. Some remain relatively localized whereas others spread globally, and their rates of transmission vary. New strains or variants of infectious diseases can evolve. These variants can alter the rates of transmission of these diseases and change their effects on morbidity and mortality. For example, the Delta strain of COVID-19 has spread more quickly and more easily than earlier strains of this virus. These uncertainties make it very difficult for businesses to predict the consequences for them of epidemics and to be financially and strategically well prepared to respond to epidemics. Furthermore, in cases where vaccines are available or treatments for infectious diseases exist, these can become less effective with the passage of time as organisms, causing these diseases to become resistant to interventions (Tisdell 2015, chp. 9).

Infectious diseases in humans can be spread by human–human contact or by vectors other than humans, such as some species of mosquitoes. In the case of human–human contact, the methods of transmission also vary. For example, there is air-borne transmission of coronaviruses as a result of human interaction but not in the case of AIDS. Transmission mechanisms influence the prospect of controlling epidemics by regulating or changing human behavior or by managing other means of transmission. In turn, the economic consequences for businesses depend on the nature of these prospects.

The risks to humankind and to their economic activities of infectious diseases have increased in the modern era due to increased urbanization as well as a greater ease of travel. Increased urbanization has resulted in concentrated population densities and in more human–human contact than in the past. This is favorable to the occurrence and spread of infectious diseases that depend on human–human contact, particularly those spread through particles in the air and germs deposited on surfaces touched by many individuals. Global and greater long-distance travel by more individuals than ever has also accelerated the rate of geographical speed of infectious diseases. Outbreaks of new infectious diseases can become widespread before this is detected or their prevalence is well known, and before the extent of their actual or potential adverse economic and other negative consequences are realized.

On the other hand, scientific advances have made it quicker to detect new diseases and to develop remedies to combat them. However, developing remedies (such as new vaccines) and ensuring their acceptance still takes time. While the development of vaccines to combat COVID-19 has been relatively rapid, many businesses have suffered substantial economic and financial losses as a result of this global pandemic and because of lags in the supply of new vaccines as well as the reluctance or unwillingness of a significant proportion of some populations to be vaccinated.

## 4. Government Policies and Business Risks

Government policies in reacting to pandemics and similar events can moderate or add to the financial risks faced by businesses. In the case of COVID-19, lockdowns, regulations requiring social distancing and the wearing of masks, as well as limitations on travel can both reduce the economic vulnerability of some businesses to the pandemic and increase that of other businesses. Early lockdown and the adoption of the other measures mentioned above once an outbreak is detected is likely to be favorable to the control of the virus and to businesses as a whole. Government grants for businesses and employees most adversely affected financially by the pandemic and by government regulations to control its spread also reduce the economic vulnerability of businesses to its occurrence (Cirera et al. 2021). On the other hand, it is difficult for firms to predict what public policies will be adopted for responding to pandemics. In fact, Iyke (2020) provide econometric evidence that the COVID-19 pandemic led to a statistically significant rise in economic policy uncertainty (EPU) in China and Korea. He states: "Although we do not find evidence that the pandemic induced economic uncertainty in India, Japan, and Singapore, the visibly high levels of the EPU indexes in 2020 cannot be a mere coincidence" (Iyke 2020, p. 3). Furthermore, the policy responses of governments to COVID-19 in different countries and jurisdictions have varied considerably. Furthermore, a study by Cirera et al. (2021) found that, on the whole, government policies to provide financial support to businesses during the COVID-19 pandemic have been poor.

The basic economic argument in favor of government intervention to control a pandemic, such as COVID-19, is that the external benefit or spillover benefit from its control are much higher than the benefits obtained by individuals. This can be illustrated in principle by Figure 1. The *X*-axis represents the percentage of a population complying with a control measure such as being vaccinated against COVID-19. Line ABC represents the marginal perceived value to individuals of being vaccinated and line ADF indicates the social marginal value of individuals being vaccinated. In the case shown, only $x_1$% of individuals will choose to be vaccinated if they are free to choose. However, this is sub-optimal from a social economic point of view. In the case, a 100% vaccination rate would be socially ideal. On the other hand, if ADF is steeper and meets the *X*-axis at $x_2$, then vaccinating $x_2$% of the population would be socially ideal, for example, if $x_2$% establishes herd immunity. In practice, the percentage of a population that needs to be vaccinated to establish herd immunity to a contagious decision is often uncertain. We still do not know what level of vaccination coverage is required to establish herd immunity to COVID-19, and the estimation of this is made more precarious as new variants of this disease evolve. Nevertheless, in

aggregate, businesses are likely to benefit from effective vaccination programs supported by governments. Government intervention helps to overcome market failures associated with private responses to pandemics and similar events.

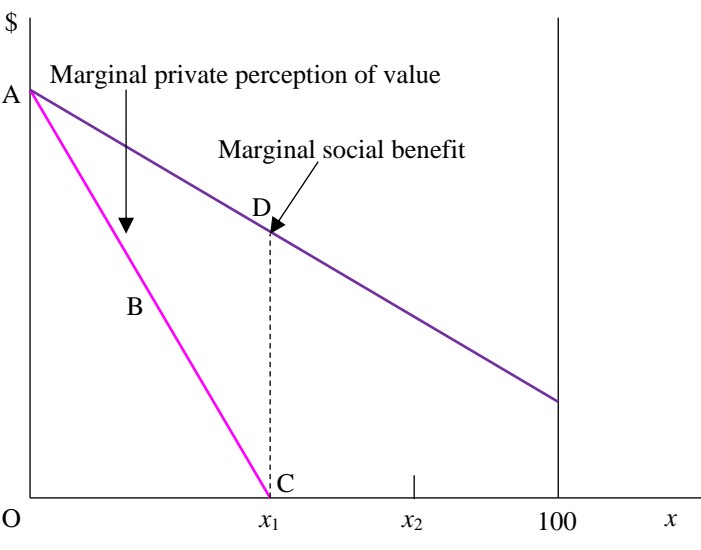

**Figure 1.** Illustration that the marginal private perception of the value of vaccination to control a pandemic can be a lot less than the marginal social benefit of vaccination. This can support government measures to counteract this problem.

The line ABC can extend below the *X*-axis because some members of the population oppose vaccination (or other measures) to control the spread and severity of infectious diseases. Some are opposed to being vaccinated with any type of vaccine; some are concerned about the possible side effects of particular vaccines and still others believe that vaccination is unnecessary. Significant reasons for COVID-19 hesitancy are outlined in Bratu (2021); Bailey et al. (2021); Davis (2021); Lăzăroiu et al. (2021); Ljungholm (2021). Furthermore, other control measures, such as the wearing of masks and social distancing (as in the case of COVID-19), may be opposed on the grounds that they are an unwarranted interference with the exercise of personal liberty (Tisdell 2020).

The use of vaccination passports may also be opposed on similar grounds. However, the larger the external benefits from complying with a control measure (that is, the larger are the external disbenefits from not doing so), the stronger the social case (other things being held constant) for restricting personal freedom in order to increase public welfare. In general, well-designed control measures which limit personal freedom in order to control pandemics benefit businesses as a whole and reduce their economic vulnerability to infectious diseases.

Lockdowns are additional control measures that reduce personal liberty. Businesses are quite vulnerable to long lockdowns where their employees cannot work effectively from home or if they rely on the physical presence of buyers for sales. Early lockdowns (and the adoption of other control measures) as soon as outbreaks are detected (and early detection of outbreaks) are likely to be beneficial for businesses as a whole. Although delays in adopting measures to control infectious disease reduce immediate costs imposed by them on businesses, delays in imposing lockdowns add substantially to subsequent economic costs because these result in these diseases becoming more widespread and difficult to manage.

Let us now consider specific attributes of businesses which make them vulnerable to the occurrence of epidemics. These include the extent to which the demand for their products or services declines and result in lost revenue; the ability of firms to adjust their

costs to cope with reduced demand for their product; the extent to which the supply of their factors of production is adversely applied; and their liquidity constraints.

## 5. Demand-Side Vulnerability

Businesses that rely for their trade on personal contacts and the physical presence of groups of people are likely to suffer a significant decline in custom and in revenue as a result of pandemics, especially if the risk of being infected by the disease is high in such circumstances, the morbidity and mortality consequences of infection are high and individuals have only limited ability to protect themselves against the disease. The reduction in demand may come about because (1) individuals are unwilling to expose themselves to the risk of being infected by personal contacts and (2) government regulations that restrict the entry of individuals to designated venues. These restrictions can include limiting the number of persons in shops and other businesses or lockdown of shops and similar businesses. Businesses involved in tourism (Gössling et al. 2021), catering, and the transport of passengers (such as airlines transporting passengers); restaurants; and businesses that rely on the presence of individuals for sales are directly vulnerable to these effects.

However, the modern economic system is an interdependent one. Consequently, many businesses that do not directly suffer a fall in demand due to an epidemic (such as COVID-19) can individually experience a serious decline in the demand for their production. In general, these firms are in those industries that have high forward economic linkages with firms in industries that experience large direct declines in the demand for their production as a result of a pandemic. Suppliers of specialized high-quality food products for restaurants and for catering businesses have experienced this problem as a result of COVID-19 (Nicola et al. 2020). In the early stages of the COVID-19 epidemic, suppliers of energy resources also experienced a significant fall in the demand for their products due to less travel and a lower level of economic activity (GDP) in many countries (Mead et al. 2020). These effects can be expected to ameliorate once infections attributable to the pandemic decline, for example, mortality and morbidity risks are generally diminished due to effective vaccines.

Padhan and Prabheesh (2021) provide empirical evidence of the major decline in oil production and the price of oil in 2020 as a result of the effect of COVID-19 and imply additional references addressing this subject. Tisdell (2021) noted a decline in energy resources used for electricity and for heat production in the same period.

Some businesses have been able to moderate the decline in the demand for their production caused by the occurrence of COVID-19. This involves relying on increased purchases of their products online, by greater reliance on contactless delivery of products, and the provision of take-away food rather than the provision of meals in-house. Small businesses can find it difficult to switch to these alternative forms of trading and they are not perfect substitutes for the supply of many commodities on site, for example, in restaurants. Feng (2020) found that businesses of this type accounted for a high proportion of declared bankruptcies in China in the early stages of the COVID-19 pandemic.

## 6. The Ability of Businesses to Adjust Their Costs of Production to Match Reduced Demand for Their Output

Businesses that suffer a significant fall in demand for their production as a result of a pandemic are endangered financially if they are unable to reduce their costs in order to avoid substantial losses. This is a serious problem for firms that have a high level of inescapable costs, such as fixed-term rental agreements, or are liable for substantial interest payments on loans.

Figure 2 illustrates a relevant case. It supposes two price-taking firms faced by identical prices for their product. They differ only in that Firm 2 has a higher level of inescapable costs than Firm 1. The total cost curve of Firm 2 is shown by the curve marked EFG and that of Firm 1 is indicated by curve ABD. The inescapable costs of Firm 2 are equivalent to the distance OE and those for Firm 2 are equal to the distance OA. Consequently, Firm 2's escapable cost exceeds those of Firm 1 by an amount equal to AE and their total cost

curves are parallel. Therefore, the firm-level characteristics of these two firms are only due to the difference in the levels of their inescapable costs.

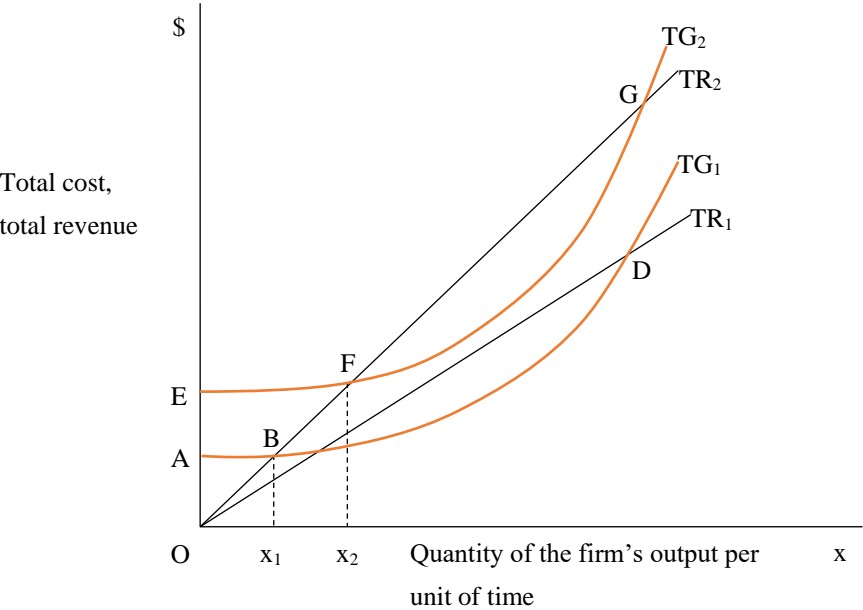

**Figure 2.** A microeconomic illustration that higher levels of inescapable costs increase the vulnerability of businesses to pandemics (such as COVID-19) if the demand for their product declines or if they are forced to reduce the supply of their product.

Let the ray marked $OTR_2$ represent the total revenue of both firms in the absence of a pandemic. The slope of the line indicates the price of their product X. They will find it most profitable to produce the same level of output and both make a profit, even though Firm 2's profit is lower than that of Firm 1 by an amount equal to AE. The interior profit maximization situation occurs for both firms.

Now suppose that, due to a pandemic, the price of the product falls and the total revenue ray for each firm becomes $OTR_1$. Firm 1 is still able to make a profit, but this will be reduced and so also will its profit-maximizing output be lowered. However, Firm 2 will be unable to break even and its profit-maximizing strategy would be able to cease operations, leading to a total financial loss of OE. Note that the break-even point for Firm 2 (prior to the pandemic) corresponds to point F and for Firm 1 to B. The corresponding break-even levels of output are $x_2$ and $x_1$, respectively.

The possibility also exists that, due to government restrictions following the occurrence of a pandemic, some businesses will have a forced reduction in their volume of sales. In these circumstances (unless the price of the product were to rise sufficiently), Firm 2 would still be more vulnerable financially to the pandemic than Firm 1. If, for example, in Figure 2, the price of a product remained unchanged (at its pre-pandemic level), a restriction on sales of $x < x_2$ would make Firm 2's business unprofitable but Firm 1 would be able to withstand a higher level of reduction in its volume of sales. It will only become unprofitable if $x < x_1$.

Note that the type of economic modelling outlined has some limitations because it is static. Furthermore, it does not allow for the possibility that the period for which costs are inescapable can differ. Some of these costs are liable to be escapable with differing durations. Moreover, the possibility of having these costs reduced as a result of negotiations has not been explored. Some businesses may also be able to obtain loans to tide them over during a pandemic following their losses; however, their ability to do so requires investigation.

It should be noted that, even if some costs are escapable, they may result in long-term economic penalties. Although it may be possible for businesses to reduce staff numbers or to lower the hours of work of staff, or to substitute lower paid staff for some of their existing

staff, businesses may no longer be financially viable if this is the case, or the quality of their product can suffer if substantial changes of this type are made. Australian universities have been heavily dependent on overseas students for generating their revenue and COVID-19 diminished the supply of these students considerably. Various risk management policies were adopted. For example, some universities in Australia have employed some lower paid (casual) staff to replace highly paid full-time ones declared to be redundant or who have been given early retirement because of reduced enrolments by students for overseas, especially China. In addition, escapable costs (such as departmental financial support) to support the research of staff members were slashed. These measures could possibly reduce the academic performance of these institutions. In addition, several of the risk-management issues for COVID-19 adopted by Chinese universities have also arisen in Australian universities.

Furthermore, if skilled staff are dismissed, it may be difficult to replace those dismissed by others with adequate skills once business conditions improve (Svizzero and Tisdell 2002). Therefore, some businesses may strive to retain a core of skilled staff (especially those with skills specific to the particular business) so that these are available when the demand for the production rises. For example, during the COVID-19 crisis, some airline companies retained a core of pilots on a part-time basis and kept them trained so they would be available when the demand for travel by air increased.

## 7. Problems in the Supply of Factors of Production

Some epidemics create problems for businesses in accessing their supplies of factors of production. This has proven to be a major problem arising from the occurrence of the COVID-19 pandemic. If employees are not able to work from home, lockdowns and the absence of infected employees from their workplaces reduce the productivity of the affected businesses (Gupta et al. 2020). Nemteanu and Dabija (2021) found that, on the whole, measures to reduce the incidence of COVID-19 resulted in reduced job satisfaction and counter-productive work behaviour. As a result, productivity declined in many Romanian companies and other institutions. Restrictions on the permitted interaction of employees (intended to reduce the incidence of contagion) can also lower the productivity of affected businesses. As pointed out by Aum et al. (2020), jobs have also been lost following lockdowns occasioned by the presence of the COVID-19 pandemic.

In some countries, some businesses rely heavily on short-term migrant labor. This is so, for example, in the Australian horticultural sector. Overseas workers on short-term visas play an important role in harvesting horticultural crops in Australia. Prohibition of the entry of these workers to Australia in the early stages of the COVID-19 pandemic caused an economic crisis for many Australian horticulturalists who were unable to harvest all of their crops due to labor shortages. Entry requirements for these workers were subsequently relaxed but these migrant workers were then subject to 14 days of quarantine after their arrival in Australia. The geographical movement of seasonal farmworkers within Australia has also been disrupted by outbreaks of COVID-19. Faulks and Yinghua (2021) also noted that in the EU the coronavirus resulted in labor shortages and demand changes as well as disturbances in farmers' supply chains. Furthermore, the supply of fly-in fly-out workers in the Australian mining industry has been disrupted by COVID-19 outbreaks.

Supply chains can be broken or delivery times can become longer as a result of pandemics, as is evident from the COVID-19 pandemic. These problems appear to me more serious during the early stages of most epidemics than in their later stages. Meier and Pinto (2020) provide some qualitative evidence which supports this hypothesis. For a period of time during the start of COVID-19, some manufacturers were short of important components for completing their manufacturing. For example, this created serious economic problems for manufacturers in the United States and in Japan who depended on the input of components from China (Meier and Pinto 2020; Zhang 2021). As China's economic activity recovered from the onslaught of COVID-19, this problem eased. This problem was worsened initially by a significant reduction in airline flights. Reduced airline

flights lowered the available capacity for carrying high value (low weight) items and perishable ones by air. Subsequently, this constraint has eased because the number of air cargo flights increased. Additionally, in some cases, local substitutes for imported ones were subsequently found, and some items that were previously air freighted were accessed by alternative means of transport, e.g., by ship.

Delays in the supply chain have resulted in an increase in stocks of components and parts (and similar items) needed down the supply chain. In Australia, delays in securing replacement parts for imported farm machinery became an economic problem during the COVID-19 pandemic. These delays prevented the timely planting and harvesting of crops. In some cases, they resulted in a substantial reduction in the incomes of affected farmers.

The above factors add to the cost of production of affected businesses and/or reduce their ability to supply their products. They result (on the whole) in an increase in the price of their products. In Australia, the occurrence of COVID-19 resulted in a significant rise in the prices of fruit and vegetables, the demand for which is relatively inelastic both in terms of price and income elasticity.

## 8. The Liquidity of Businesses and Their Vulnerability to Pandemics

Most economic theories—for example, those outlined in microeconomic texts as well as, such as that of Hicks (1946)—pay little or no attention to the liquidity of businesses as an influence on their survival. However, firms having a high degree of liquidity are more likely to survive adverse economic conditions than those lacking in liquidity.

The liquidity of business depends on several factors. These include their amount of cash on hand and other assets which are easily converted to cash, such as investments in low-risk securities. If a business has the ability to access loans (e.g., bank loans), this can provide it with an increase in liquidity for a limited period of time. Bartik et al. (2020, p. 17656) found that, as a result of a survey of United States small business, many small businesses were "financially fragile. The median business with more than $10,000 in monthly expenses had about 2 weeks of cash on hand at the time of the survey" conducted in March and April, 2020, i.e., during the early stages of the COVID-19 pandemic. Raising extra capital by means of new share issues is also a possibility; however, in a depressed market caused by a pandemic, this can be difficult for those businesses adversely affected by a pandemic.

Factors that can strain the liquidity of firms during pandemics (apart from their reduced revenues) include an increase in the level of their bad debts and slower payment or non-payment of debts by buyers. Businesses with high gearing ratios and high levels of inescapable costs of production are especially prone to liquidity crises if they are adversely affected by a pandemic.

It is pertinent to note that it can be economically difficult for businesses to adopt policies that increase their prospects of being able to withstand the adverse economic consequences of a pandemic. For example, the occurrence of epidemics, their consequences, and deduction can be very uncertain, especially a novel one, such as COVID-19 (Ma et al. 2020). Ma et al. (2020) state that "much of the uncertainty concerning the ultimate economic and financial effects from the COVID-19 stems from the unknown timing and severity of the virus". Furthermore, maintaining greater liquidity to cope with a possible epidemic usually involves an opportunity cost because the funds involved can often earn a higher return if invested in less liquid assets. Consequently, if an epidemic does not become widespread and last long, income is foregone by businesses if they increase their liquidity because they predict a longer and more widespread epidemic than that which actually emerges. The above conceptual results identify firm-specific business characteristics which make firms vulnerable to the effects of pandemics. They are consistent with the empirical findings of Xiong et al. (2020) about how firm-specific characteristics affected the vulnerability of corporations in China during the early stages of the COVID-19 outbreak.

## 9. Results and Discussion

Results arising from the microeconomic analysis of the factors affecting the economic vulnerability of businesses to pandemics (such as COVID-19) have been reported in the above sections of this article. The most important ones are summarized in Figure 3. They can adversely affect the demand, supply, and cost conditions facing individual businesses. Some empirical examples are provided in this section. These are in addition to those given elsewhere in this article.

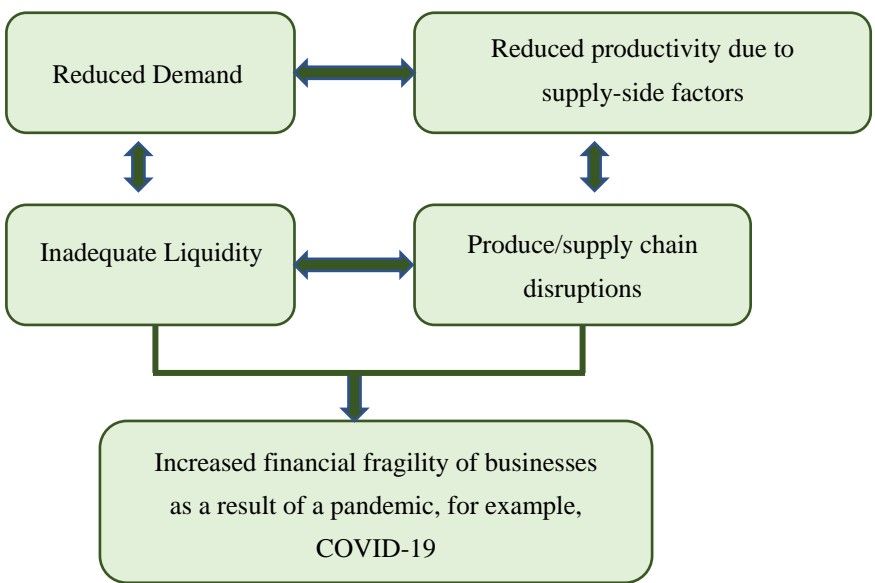

**Figure 3.** Pandemics can result in all the economic effects stated above and thereby increase in the financial vulnerability of potentially affected businesses. These effects are often independent.

Note also that differences in the nature of pandemics and government policies for dealing with these are consequential (as was shown in Sections 3 and 4) for the vulnerability of businesses to pandemics. These are not highlighted by Figure 3 but might be shown in extra cells.

Epidemics vary considerably in their nature, duration, geographical spread, rapidity of spread, and economic consequences. These factors and their occurrence are difficult to predict in advance. This limits the ability of businesses to be prepared for them. Nevertheless, it is predicted that epidemics will become more frequent due to high levels of human population; greater human interaction, as a result of higher densities of population in cities; and greater mobility of individuals. This article has identified a set of factors that make individual businesses economically vulnerable to pandemics and similar events. This vulnerability depends on the nature of an infectious disease and the type of government policies adopted to ameliorate their adverse economic effects. Relevant government policies include financial assistance to firms adversely affected by an epidemic and measures to limit the spread and occurrence of the relevant disease.

It has been demonstrated that businesses are especially vulnerable to pandemics if these significantly reduce the demand for production and if they have a relatively high level of inescapable costs. Businesses in tourism, catering, and on-site entertainment are at particular risk. However, suppliers of inputs to these industries also can experience a slump in their revenues. The vulnerability of businesses to epidemics can be exacerbated by delays or breaks in product chains, particularly those chains that are international in nature. Furthermore, restrictions on the geographical movement of labor (introduced to reduce the spread of infectious disease) can seriously disrupt the production and profitability of some businesses and add to the costs of their production. This group of businesses include those relying on seasonal migrants to assist with their production as do (for example) several Australian agricultural industries. Again, a recent intervention by the New South

Wales Government to address this included offering public servants five days' paid leave to reduce those labor shortages on farms (The Guardian 2021). It was also stressed that in assessing the economic vulnerability of businesses to epidemics that their liquidity needs to be taken into account and that consideration should be given to the possibility of their becoming insolvent as a result of an increase in the level of the bad debts. Further discussion on the economic vulnerability of businesses to uncertain events is available in Tisdell (2013).

The vulnerability of business in different types of industries caused pandemics such as COVID-19 was given some consideration. Schoenfeld (2020) gave an *early assessment* of this for COVID-19 based on the change in market value of companies listed on the New York Stock Exchange. Businesses recording the largest decrease in their value (based on their changed share price) included restaurant and hotel firms, transportation firms, aircraft, and ship and railroad firms. Companies operating cruise lines experienced a decline in value of 80 per cent. Rises were recorded for Walmart, General Mills, Netflix, and some pharmaceutical companies, presumably because the demand for their output was not expected to decline. These changes, according to Schoenfeld, basically reflect predicted changes due to COVID-19 in the level of demand for the output of the companies mentioned. This is consistent with the analysis in Section 4 of this article.

As pointed out by one of the reviewers of this article, many businesses that experienced a significant decline in their volume of sales due to the COVID pandemic were able to adjust successfully to the event in a variety of ways. Some did this by reducing their escapable costs (for example, lowering their labor cost by diminishing the amount of labor employed) and some were able to defer their inescapable costs (for instance, defer loan repayments or negotiate a slow rate of repayment of loans). This reviewer also pointed out that the earnings of many firms before interest and tax (EBIT) remained positive (albeit at a low level) despite a substantial decline in their revenue. Depending on the type of business, the possibility emerged because they were in a position to cut costs fairly rapidly or could limit the decline in their sales by adopting new initiatives, for example, offering home delivery of products or, in the case of restaurants, making take-aways available.

It is not possible here to provide a comprehensive coverage of the financial effects and the ways in which different individual businesses coped with the COVID pandemic. However, as pointed out above, the tourism and travel sector were hard hit financially by the pandemic. Nevertheless, many businesses in the sector survived, often as a result of government aid but not always. It has been predicted that the earnings of those surviving businesses will rise substantially in 2022 and 2023 as the pandemic is controlled. In Australia, for example, the travel agency Flight Centre (listed on the Australian Stock Exchange) survived despite a massive reduction in the value of its sales. It was able to reduce its labor costs by diminishing staff numbers and by closing many of its offices. Delays in refunding prepaid tourism/travel bookings also helped to maintain its liquidity. Furthermore, emphasis was placed on promoting domestic tourism rather than international tourism. Similar shortages were also adopted by airlines which, as is well known, experienced a huge reduction in their revenue in 2020 compared to 2019 because of COVID-19 and because of government measures to reduce infections. Most airlines received government financial assistance in order to remain viable. However, they could not rely on this assistance alone to survive. The strategies which they adopted to cope with the situation are well outlined and reviewed in Albers and Rundshagen (2020).

Some companies, on the other hand, prospered financially as a result of COVID-19. For example, the profits of Zoom Inc rose from USD 21.7 million in 2019 to USD 671.5 million in 2020 (BBC News 2021) due to the burgeoning demand for the use of its software video conferencing for businesses and education. Similarly, the net income of Amazon rose from USD 11,588 million in 2019 to USD 21,331 million due to its online business activities. Although not all of these increases may be due *entirely* to the occurrence of COVID-19, COVID-19 significantly accelerated the demand for the services supplied by these businesses.

Changes in the relative value of companies (as per stock market variations) during the COVID-19 pandemic have varied with the passage of time. In the early stages of the COVID-19 pandemic, share prices fell considerably as a result of panic selling and the prospect of selling and repurchasing shares when their value bottomed or nearly so. Panic selling subsided later and (overall) share values recovered. These time-related changes are worthy of more in-depth study. A second matter requiring consideration is the extent to which the value of shares in companies accurately reflects the net present value (NPV) of companies. Because of speculation and other factors, they can be a poor guide to the NPV of companies.

However, it is pertinent to note that scholarly hypothesis about the efficiency of financial markets differ (Tomer 2017, chp. 6). Malkiel (1999); Shleifer (2000) have argued that the value of stocks closely mirrors their NPV. On the other hand, Shiller (1981, 2005) rejects this hypothesis for several reasons. In my view, the early adjustment of the prices of stocks to the occurrence of COVID-19 did not reflect well their long-term value, and, therefore, was at odds with the behavioral view that financial markets are extremely efficient. Consequently, the observations of Schoenfeld (2020) may not precisely reflect the change in the NPVs of corporations mentioned. Overall, the adverse economic consequences of this pandemic have outweighed the economic gains to particular businesses (Ma et al. 2020; Kose et al. 2020). While increased expenditure on healthcare due to the occurrence of COVID-19 has added to the value of GDP, it would be more appropriate to consider this as a cost rather than an economic benefit, as further iterated by Nordhaus (1969); Tobin (1964) in their critiques of relying on GDP values as a measure of aggregate economic welfare. They suggested that a new economic welfare (NEW) concept should be adopted. Similarly. Barkley and Seckler (1972) proposed net social welfare (NSW) as a more relevant economic welfare indicator. They point out that expenditures resulting from ill health and adverse environmental conditions can add to the level of GDP but that those ought to be regarded as costs of these phenomena, not benefits. Similarly, *some* of the economic effects of COVID-19 add to GDP (such as extra spending on medicinal treatments and burials) but these should not be regarded as economic benefits. Furthermore, economic gains by some types of businesses as a result of COVID-19 (e.g., debt collection firms, psychiatrists) add to GDP but are actually negative effects of this pandemic. Therefore, the negative economic consequences of COVID-19 have been more adverse than those indicated by declines in GDP.

## 10. Conclusions

This original article has systematically identified the general economic characteristics of individual firms that make them vulnerable to the consequences of pandemics, for example, COVID-19. The conceptual analysis has been mainly based on the microeconomic theory of the production of a business. The theoretical analysis has been linked to empirical observations in the relevant literature where available and the relevant literature has been reviewed. The literature is consistent with the analysis but also considers some relevant aspects not covered by the above theory. For instance, the role which organizational attributes can play in reducing the vulnerability of business to pandemics is not covered but is explored, for instance, by Obrenovic et al. (2020) and differences in business perceptions about the likely negative economic consequences of COVID-19 and its severity have not been examined. A survey by Bartik et al. (2020) found that the expectations of small business in the latter respect varied considerably. Therefore, this article only covers a part of the factors that influence the vulnerability of individual businesses to pandemics, albeit an important part. It also provides a guide to factors which reduce the financial risk of firms from exposure to pandemics.

Some businesses during the COVID-19 pandemic have been able to survive (and even prosper) due to their superior entrepreneurship. Some have been able to take advantage of increased online sales or services involving minimal human contact. However, this has not been possible for all businesses, for example, those in which personal contact is

essential. Some businesses have made greater use of robotics and automation to overcome labor shortages or plan to do so. Others have found ways to limit or avoid product chain disruption. These aspects should be considered in future studies of this subject.

**Funding:** This research received no external funding.

**Acknowledgments:** I would like to thank the four anonymous reviewers for their constructive feedback on this article. I would also like to thank Evelyn Smart for her research help and production input. The useful comments of Hemanath Swarna Nantha on the previous version of this article are appreciated. The usual caveat applies.

**Conflicts of Interest:** The author declares no conflict of interest.

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
