# Peer review of "Determinants of the Economic Vulnerability of Businesses to Pandemics and Similar Events"

_jrfm, doi:10.3390/jrfm14110532_

Round 1
Reviewer 1 Report
I think that the paper is very well written and reflects a proper view point of a well established and well known scholar. I only think that these view point could be enhanced by adding some more references from the international literature regarding businesses and COVID-19. It would also be good if the reference list would be formatted properly, as for 2 references it is not clear who are the authors.
There are a lot of unreferenced passages. For example, on COVID-19 vaccine risk beliefs, perceptions, attitudes, and intentions you might like to cite such Scopus Q1 sources:
Lăzăroiu, G., Mihăilă, R., and BraniÈ™te, L. (2021). “The Language of Misinformation Literacy: COVID-19 Vaccine Hesitancy Attitudes, Behaviors, and Perceptions,” Linguistic and Philosophical Investigations 20: 85–94. doi: 10.22381/LPI2020217.
Bratu, S. (2021). “Individual and Psychosocial Features Shaping COVID-19 Vaccine Hesitancy: Attitudes, Beliefs, and Norms,” Linguistic and Philosophical Investigations 20: 125–134. doi: 10.22381/LPI20202111.
Bailey, L., Grupac, M., and Sosedova, J. (2021). “COVID-19 Vaccine Risk Beliefs, Perceptions, Attitudes, and Intentions,” Review of Contemporary Philosophy 20: 81–92. doi: 10.22381/RCP2020214.
Davis, R. (2021). “COVID-19 Vaccine Hesitancy, Delay, and Refusal: Insufficient Knowledge, Complacency, and Distrust of the Medical System,” Review of Contemporary Philosophy 20: 139–150. doi: 10.22381/RCP2020219.
Popescu Ljungholm, D. (2021). “COVID-19 Threat Perceptions and Vaccine Hesitancy: Safety and Efficacy Concerns,” Analysis and Metaphysics 20: 50–61. doi: 10.22381/am2020213.
Priem, R. (2021). “An Exploratory Study on the Impact of the COVID-19 Confinement on the Financial Behavior of Individual Investors,” Economics, Management, and Financial Markets 16(3): 9–40. doi: 10.22381/emfm16320211.
Nemțeanu, M.S., and Dabija, D.C. (2021). "The influence of internal marketing and job satisfaction on task performance and counterproductive work behaviour in an Emergent Market during the COVID-19 pandemic". International Journal of Environmental Research and Public Health, 18(7): 3670. doi: 10.3390/ijerph18073670.
Author Response
Thank you for a positive report and your constructive suggestions which have been attended to. All of your suggested references have been included in the revised version of this article together with many others.
Reviewer 2 Report
- The research hypothesis is not clear: it is very difficult to understand the goal of the whole proposed paper and to evaluate if the author finally achieve it;
- The research methodology completely is missing: it is difficult to check if the author's conclusions are sound & clear from a scientific point of view;
- The paper is more general and less specific: I suggest improving the research by adding examples, data, charts, and figures that can support the whole research;
- The theoretical background of the paper is very poor: I suggest looking to the economic theory of crisis & business cycles, to the solutions, and, especially, to the state intervention during the crisis. A taxonomy of crisis nowadays and a deep look at their nature could help a lot.
- The conclusions are too general, not properly connected with previous studies;
- The limitations of the research are not mentioned;
- The solutions & recommendations are missing and not clearly assumed by the author.
Author Response
Thank you for taking the time tor review this article. It has been extensively revised to make the goal of the article clearer and to point out that in terms of methodology, its main contribution is as a conceptual or theory paper based on the microeconomics of the firm. It also includes empirical results drawn from the relevant literature.
The paper has been made more specific by adding figures and charts and examples are provided. The theoretical background of this paper is not based on macroeconomics, e.g. the theory of crises and business cycles. This would be a different article. Several, but not many articles in the relevant literature do consider the firm-specific attributes of businesses as an influence on vulnerability to pandemics including COVID-19. However, they do not cover the economic theory of this in the same systematic manner in this article. The microeconomics of this is made clearer in the revision by use of Figures 2 and 3 and the surrounding text. Considerably more attention has been given in the revision to linking this article with findings in related articles.
Both a discussion and concluding section are now present in the revised article. These should address remaining concerns and particular attention has been given to the limitations of this research. Thank you for your helpful suggestions.
Reviewer 3 Report
The paper covers a very topical issue "vulnerability of businesses to pandemics". However, the paper does not meet the commonly accepted requirements to be published in JRFM.
- The author does not follow IMRAD structure
- The literature review is vague; it is based on 15 sources in total. Each sub-chapter contains only 1-2 references.
- There is neither empirical research nor review of the literature. The paper looks like an essay. The scientific contribution of the author is not clear.
- Conclusions are not supported by research results because there is no research with clearly described results.
Author Response
Thank you for taking the time to review this article. Regarding your specific points:
- The structure of this article has been improved as a result of the revision, and its formatting accords with the requirements of JRFM.
- The number of references has been substantially increased and reviewed in the appropriate places.
- The contribution of this article is of a conceptual/theoretical nature and based on the microeconomic theory of the firm. This is the original scientific contribution. The theory is linked with empirical results drawn from the relevant literature wherever possible.
- The revised version of this article does support the results of the research as a consequence of the changes to and substantial additions to the text.
Reviewer 4 Report
After making the requested modification the paper will add to the existing body of knowledge on the topic. In order to do so, the authors should improve the literature review, and discussion section, introduce future studies paragraphs.
- The introduction section does not succeed in putting the current research in the context of the existing literature, thus illustrating the significance of the study. Overall originality and contributions should be emphasized in the introduction better and connected with literature gaps.
- The overall logic and coherence of the paper is of a good quality, but theoretical review should be richer. Poor literature review with around 20 article sources cited. The paper should contain at least 50 references. Add more references and also, review new articles published in 2020 and 2021.
Some published article deal with same or similar topic:
Obrenovic, B., Du, J., Godinic, D., Tsoy, D., Khan, M. A. S., & Jakhongirov, I. (2020). Sustaining enterprise operations and productivity during the COVID-19 pandemic:“Enterprise Effectiveness and Sustainability Model”. Sustainability, 12(15), 5981. Saif, N., Ruan, J., & Obrenovic, B. (2021). Sustaining Trade during COVID-19 Pandemic: Establishing a Conceptual Model Including COVID-19 Impact. Sustainability, 13(10), 5418.The authors should put more effort to review literature that has conceptualized or already empirically tested the same relationships.
Author Response
Thank you for your very constructive review. Your suggested modifications to the article have now been made and the new version is much improved. The following are some of the changes.
- The introductory section has been revised in the manner suggested by you.
- The exposition of the theoretical content of the article has been improved by adding to the text and by adding a couple of extra figures. The number of references has been substantially increased and their relevance made clearer. The two references suggested by you have been added.
- Greater effort has been made to review the literature that has conceptualized or empirically tested the same relationships as those considered in this article.
The above have all resulted in a significant improvement in this article.
Round 2
Reviewer 2 Report
I confirm that this version of the paper is significantly improved than the previous one.
However, I suggest to add some empirical evidences at the end of your study. The paper that you are proposing remains too simplistic and merely theoretical. Some data about various economic sectors could be valuable. For instance, we noticed that, during the pandemic, the sales volume significantly decreased but EBIT remained positive (small value but positive), due to significant business adjustments made by the companies during this hard time, especially for the SMEs. Moreover, not all economic sectors have been affected by pandemic crisis in the same way. Some of the sectors adjusted the labor costs, others adjusted the production costs etc. with a direct impact on the liquidity. However, the fueling with cheap liquidity seems to have also a negative / opposite inflationary impact these days, all over the world (the energy price, the fuel price etc.).
I think that it will be more valuable for your paper to add some data or examples that confirm the very interesting Figure from the Discussion section of the paper. There are data now available for 2020 that can show the drop of the demand, the worsening of liquidity in the private businesses and so on.
Reviewer 3 Report
The paper has a conceptual/theoretical nature. The authors provided critical discussions of the previous studies, and their viewpoint is based on the analysis of the substantial volume of the literature.
Major comments:
- The authors partly followed the recommendations to base the structure on IMRAD. The parts "Methodology" and "Results" are missing. It disturbs in understanding what exactly is the authors' contribution
- The goal of the paper is to "...identify .... factors", but there is no any scheme/figure/table with the summary of the identified factors. Figure 3, for instance, contains risks as a result of COVID-19, but it was not the purpose. Again, that is why, the separate part "Results" is absolutely necessary.